# Unraveling the Mechanisms of Sensitivity to Anti-FGF Therapies in Imatinib-Resistant Gastrointestinal Stromal Tumors (GIST) Lacking Secondary *KIT* Mutations

**DOI:** 10.3390/cancers15225354

**Published:** 2023-11-09

**Authors:** Sergei Boichuk, Pavel Dunaev, Vera Skripova, Aigul Galembikova, Firyuza Bikinieva, Elena Shagimardanova, Guzel Gazizova, Ruslan Deviatiiarov, Elena Valeeva, Ekaterina Mikheeva, Maria Vasilyeva, Pavel Kopnin, Vladimir Strelnikov, Ramziya Kiyamova

**Affiliations:** 1Department of Pathology, Kazan State Medical University, Kazan 420012, Russia; dunaevpavel@mail.ru (P.D.); ailuk000@mail.ru (A.G.); firuza1995@mail.ru (F.B.); miheeva.1973@bk.ru (E.M.); 2Department of Radiotherapy and Radiology, Russian Medical Academy of Continuous Professional Education, Moscow 127051, Russia; 3Central Research Laboratory, Kazan State Medical University, Kazan 420012, Russia; vevaleeva@yandex.ru; 4Biomarker Research Laboratory, Institute of Fundamental Medicine and Biology, Kazan Federal University, Kazan 420008, Russia; vsskripova@kpfu.ru (V.S.); rgkiyamova@kpfu.ru (R.K.); 5Regulatory Genomics Research Center, Institute of Fundamental Medicine and Biology, Kazan Federal University, Kazan 420008, Russia; rjuka@mail.ru (E.S.); grgazizova@gmail.com (G.G.); ruselusalbus@gmail.com (R.D.); 6LIFT—Life Improvement by Future Technologies Institute, Moscow 121205, Russia; 7Loginov Moscow Clinical Scientific Center, Moscow 111123, Russia; 8Cytogenetics Laboratory, Carcinogenesis Institute, N.N. Blokhin National Medical Research Center of Oncology, Moscow 115478, Russia; mvnovikova94@mail.ru (M.V.); pbkopnin@mail.ru (P.K.); 9Epigenetics Laboratory, Research Centre for Medical Genetics, Moscow 115522, Russia; vstrel@list.ru

**Keywords:** gastrointestinal stromal tumor cells (GISTs), imatinib (IM), Infigratinib (BGJ 398), resistance, receptor tyrosine kinase (RTK), c-KIT and FGFR-signaling, FGF-2, autocrine pathway, clonal heterogeneity

## Abstract

**Simple Summary:**

Acquired resistance of gastrointestinal stromal tumors (GISTs) to imatinib mesylate (IM) is one of the most critical challenges in GIST therapy. Here we show that a long-term culture of GIST T-1 cells with IM induces clonal heterogeneity resulting in the appearance of cancer cells exhibiting activation of the FGFR-signaling pathway which was associated with KIT loss. The first one was due to the overexpression of FGFR1/2 and increased production of FGF-2 ligand. These events maintained GIST resistance to IM and rendered these GIST cells highly sensitive to all types of pan-FGFR-inhibitors used in the current study. Knockout of *FGFR2* in this GIST subclone significantly attenuated pro-apoptotic and anti-proliferative activities of infigratinib (BGJ 398) both in vitro and in vivo, thereby suggesting the activation of FGFR-signaling pathway via FGFR2-mediated axis as the predominant molecular mechanism in these GIST cells. Collectively, the extended inhibition of KIT-signaling in GISTs induces clonal heterogeneity of cancer cells and might change the tumor’s sensitivity to FGFR-inhibitors due to selection of cancer cells with an FGFR-overactivated pathway.

**Abstract:**

We showed previously that inhibition of KIT signaling in GISTs activates FGFR-signaling pathway rendering cancer cells resistant to receptor tyrosine kinase inhibitor (RTKi) imatinib mesylate (IM) (Gleevec) despite of absence of secondary *KIT* mutations and thereby illustrating a rationale for the combined (e.g., KIT- and FGFR-targeted) therapies. We show here that long-term culture of IM-resistant GISTs (GIST-R1) with IM substantially down-regulates KIT expression and induces activation of the FGFR-signaling cascade, evidenced by increased expression of total and phosphorylated forms of FGFR1 and 2, FGF-2, and FRS-2, the well-known adaptor protein of the FGF-signaling cascade. This resulted in activation of both AKT- and MAPK-signaling pathways shown on mRNA and protein levels, and rendered cancer cells highly sensitive to pan-FGFR-inhibitors (BGJ 398, AZD 4547, and TAS-120). Indeed, we observed a significant decrease of IC50 values for BGJ 398 in the GIST subclone (GIST-R2) derived from GIST-R1 cells continuously treated with IM for up to 12 months. An increased sensitivity of GIST-R2 cells to FGFR inhibition was also revealed on the xenograft models, illustrating a substantial (>70%) decrease in tumor size in BGJ 398-treated animals when treated with this pan-FGFR inhibitor. Similarly, an increased intra-tumoral apoptosis as detected by immunohistochemical (IHC)-staining for cleaved caspase-3 on day 5 of the treatment was found. As expected, both BGJ 398 and IM used alone lacked the pro-apoptotic and growth-inhibitory activities on GIST-R1 xenografts, thereby revealing their resistance to these TKis when used alone. Important, the knockdown of *FGFR2*, and, in much less content, *FGF-2*, abrogated BGJ 398′s activity against GIST-R2 cells both in vitro and in vivo, thereby illustrating the FGF-2/FGFR2-signaling axis in IM-resistant GISTs as a primary molecular target for this RTKi. Collectively, our data illustrates that continuous inhibition of KIT signaling in IM-resistant GISTs lacking secondary *KIT* mutations induced clonal heterogeneity of GISTs and resulted in accumulation of cancer cells with overexpressed FGF-2 and FGFR1/2, thereby leading to activation of FGFR-signaling. This in turn rendered these cells extremely sensitive to the pan-FGFR inhibitors used in combination with IM, or even alone, and suggests a rationale to re-evaluate the effectiveness of FGFR-inhibitors in order to improve the second-line therapeutic strategies for selected subgroups of GIST patients (e.g., IM-resistant GISTs lacking secondary *KIT* mutations and exhibiting the activation of the FGFR-signaling pathway).

## 1. Introduction

Gastrointestinal stromal tumors (GISTs) are the mesenchymal neoplasms of the gastrointestinal tract arising from the specialized interstitial cells of Cajal (ICCs), or gastrointestinal pacemaker cells that control gut motility. Given that GISTs are most frequently driven by auto-activated, mutant KIT receptor tyrosine kinase gene and, less commonly, by the platelet-derived growth factor receptor alpha (PDGFR-α) [1,2,3], receptor tyrosine kinase inhibitor (RTKi) imatinib mesylate (IM) (Gleevec) is currently accepted as the first-line therapeutic option for GIST patients [4,5]. However, despite the impressive response rates after initiation of IM-based therapy, more than a half of patients with advanced, non-resectable and metastatic GISTs acquire resistance to this targeted drug due to the multiple molecular mechanisms. A long time ago, secondary *KIT* mutations in GISTs were found as a predominant molecular mechanism of acquired IM resistance [6,7,8]. However, the fundamental challenge in GIST therapy is development of resistance to IM and other RTKi targeting the KIT/PDFGRA-signaling pathway due to activation of the parallel signaling molecular cascades, regulating survival and proliferation of cancer cells. Therefore, for non-resectable, recurrent and metastatic GISTs, there is an urgent need to develop novel therapeutic strategies for targeting GISTs beyond the KIT signaling axis. In particular, activation of MET/AXL-kinases associated with loss of c-KIT (named “kinase switch”) [9] was shown as an alternative molecular mechanism of acquired GIST resistance to IM. In addition, overexpression of focal adhesion kinase (FAK) [10], amplification of the insulin-like growth factor receptor I (IGF-1R) [11], and *BRAF V600E* mutation (5% GIST) [12] were also found to be implicated in GIST resistance to IM, thereby providing a rationale for clinical use of corresponding RTKi for the selected subgroups of GIST patients.

Our previous studies evidenced that an acquired resistance to IM in GISTs lacking secondary *KIT* mutations might be due to activation of the FGFR-signaling pathway [13]. Moreover, we observed that FGFR-inhibitors (e.g., BGJ 398) effectively re-sensitized GIST to IM both in vitro and an in vivo, thereby providing a rationale for the combined therapies by the aforementioned RTKi for the selected groups of GIST patients lacking secondary *KIT* mutations, exhibiting activation of FGFR-signaling pathway, and progressing during the first-line IM-based targeted therapy [14]. Lastly, we found that IM induced the autocrine-based activation of FGFR-signaling in GISTs via overproduction and secretion of the FGF-2 ligand. This was also evidenced for GIST cell lines treated with IM in vitro, GIST xenografts and clinical samples [15]. Of note, neutralization of FGF-2 abolished GIST resistance to IM and effectively abrogated the migratory and invasive capacities of cancer cells in vitro [15]. Thus, we concluded that IM induces activation of FGF-signaling in GISTs and this might have a significant impact on the malignant behavior of IM-resistant GISTs and disease progression. These results were in agreement with previous findings illustrating the activation of FGFR-signaling in GISTs as feedback of the long-term inhibition of KIT-signaling in GIST [16,17], thereby illustrating the novel mechanism of IM resistance for GIST patients lacking secondary *KIT* mutations and highlighting that FGFR inhibition might improve the long-term efficacy of IM-based therapy in GIST patients.

We showed here that the long-term (i.e., >12 months) exposure of IM-resistant GIST to IM induced clonal heterogeneity of GIST cells thereby producing a GIST subclone characterized by “KIT loss” and overexpression of FGFR1/2. This in turn abolished a synergy between IM and BGJ 398 (KIT- and pan-FGFR-inhibitor, respectively) in GIST cells which we observed earlier after a short-term (less than 4 months) culture of GIST with IM [15]. Importantly, these cells acquired significant sensitivity to the pan-FGFR inhibitors (i.e., BGJ 398, AZD 4547, and TAS-120), which was evidenced by a substantial increase of apoptotic cell death in vitro. This pattern was also revealed in vivo by examining the tumor specimens of BGJ 398-treated mice bearing IM-resistant GIST xenografts. Since none of the activation mutations in *FGFR1-4* were identified in BGJ 398-sensitive GIST, we speculated that their sensitivity to pan-FGFR-inhibitors was solely due to overexpression of the FGFR1/2, which resulted in the activation (i.e., phosphorylation) of FGFR1/2 and FRS-2, the well-known adaptor protein of FGFR-signaling. Thus, despite KIT loss, these GIST cells exhibited activation of the downstream signaling cascades maintaining cellular proliferation and survival. This was revealed by western blotting and by the comparative transcriptome analysis of GIST T-1 naive, IM-resistant GISTs (further shown here as GIST-R1) and BGJ-398 sensitive (further shown here as GIST-R2) cells, illustrating significant activation of both PI3K/Akt- and MAPK-signaling cascades in GIST-R2 cells despite the lack of KIT-signaling. To examine the molecular mechanisms of FGFR-activation in these GIST cells more precisely, we performed CRISPR/Cas9-mediated knockout of the *FGF2*, *FGFR1* and *FGFR2*. We observed that *FGFR2*−/− GIST cells were much less sensitive to FGFR inhibition both in vitro and in vivo. In contrast *FGF2*−/− and *FGFR1*−/− cells retained sensitivity to the pan-FGFR-inhibitors.

Collectively, our data illustrate that continuous inhibition of KIT signaling in IM-resistant GISTs lacking secondary *KIT* mutations raised the clonal heterogeneity of cancer cells and induced accumulation of GIST cells with overexpressed FGFR1/2, further leading to activation of FGFR- and downstream signaling pathways. This in turn rendered these cells extremely sensitive to the corresponding RTKi, used even alone, and therefore provides a rationale to further re-evaluate the effectiveness of FGFR1/2-inhibition in order to improve therapeutic strategies for patients with IM-resistant GISTs with acquired secondary resistance to IM via KIT-independent mechanisms.

## 2. Materials and Methods

### 2.1. Chemical Compounds

Imatinib mesylate (IM), infigratinib (BGJ 398), AZD 4547, PD 173074, TAS-120, H3B-6527 were obtained from SelleckChem (Houston, TX, USA).

### 2.2. Antibodies

Primary antibodies used for immunoblotting and immunofluorescence were as follows: phospho-MAPK (Erk1/2) Thr202/Tyr204 (cat. no. 4370S), MAPK (Erk1/2) (cat. no. 4696S), phospho-KIT Y719 (cat. no. 3391S), phospho-AKT S473 (cat. no.4060P), AKT (cat. no. 4691P), phospho-FRS2α Y196 (cat. no. 3864S) and Y436 (cat. no. 3861S), phospho-FGFR Y653/654 (#cat. no. 3476S), FGFR1 (cat. no. 9740S), FGFR2 (cat. no. 23328S), FGFR3 (cat. no. 4574S), phospho-STAT-1 Ty701 (cat. no. 7649T), STAT-1 (cat. no. 14994T), phospho-STAT-3 Tyr705 (cat. no. 9145T), STAT-3 (cat. no. 4904T), phospho-GSK3B Ser9 (cat no. 5558T), GSK3B (cat. no. 12456S), phospho-S6 Ribosomal Protein Ser235/236 (cat. no. 4858T), cleaved form of caspase-3 (cat. no. 9662S), cleaved PARP (cat. no. 5625) (Cell Signaling, Danvers, MA, USA), FGFR4 (cat. no. SAB1300019) (Sigma-Aldlirch, St. Louis, MO, USA), c-KIT (cat. no. A4502, Dako, Carpinteria, CA, USA), FGF-2 (cat. no. sc-365106), FRS-2 (cat. no. sc-17841), c-KIT (cat.no. sc-13508) (Santa Cruz Biotechnology, Santa Cruz, CA, USA), beta-actin (cat. no. A00730-200, GenScript, Piscataway, NJ, USA), Cas-9 (cat. no. MA1-202), *β*-tubulin (cat. no. MA5-16308-HRP), PARP (cat. no. 436400) (Invitrogen, Walthman, MA, USA). HRP-conjugated secondary antibodies for Western blotting were purchased from Santa Cruz. For IHC-staining a cleaved form of caspase-3 (#cat. no. 9662S) (Cell Signaling, Danvers, MA, USA) was used.

### 2.3. Cell Lines and Culture Conditions

GIST T-1 was established from a metastatic pleural tumor from a stomach GIST and contains heterozygous 57-base pair deletion (V570-Y578) in *KIT* exon 11 [18]. IM-resistant GIST T-1R subline was established in our laboratory after a continuous induction from 0.4 nM to 1000 nM IM in a stepwise increasing concentration manner [13]. GIST cells were grown in a humidified atmosphere of 5% CO_2_ at 37 °C (LamSystems, Miass, Russia).

### 2.4. Plasmids

psPAX2 (Addgene plasmid #12260; http://n2t.net/addgene:12260 (accessed on 5 November 2023); RRID:Addgene_12260) and pMD2.G (Addgene plasmid # 12259; http://n2t.net/addgene:12259 (accessed on 5 November 2023); RRID:Addgene_12259) were gifts from Didier Trono; pCW-Cas9 was a gift from Eric Lander & David Sabatini (Addgene plasmid #50661; http://n2t.net/addgene:50661 (accessed on 5 November 2023); RRID:Addgene_50661) [19]; pLenti-SG-1 was a gift from Dr. Igor Astsaturov (Fox Chase Cancer Center, Philadelphia, PA, USA)

### 2.5. Bacterial Transformation and Plasmid DNA Purification

XL1-Blue chemically competent *E. coli* cells (Evrogen, Moscow, Russia) were used to amplify plasmid DNA. Bacterial transformation was performed by a standard heat shock method. Plasmid DNA was purified from XL1-Blue cells using Plasmid Miniprep or Plasmid Midiprep kits (Evrogen, Moscow, Russia) in accordance with the manufacturer’s recommendations. Plasmid DNA concentration was measured using a NanoDrop Lite Spectrophotometer (Thermo Fisher Scientific, Waltham, MA, USA).

### 2.6. Lentivirus Production

Production and packaging of lentiviruses were performed in HEK293T cells by poly-ethylenimine (PEI) transfection in 12-well plates. psPAX2 and pMD2.G were used as packaging plasmids. The transfection mix was prepared in 100 µL of Opti-MEM I medium with 0.05 µg/mL PEI, 0.6 µg of transfer plasmid coding Cas9 or sgRNAs, 0.425 µg of psPAX2 and 0.25 µg of pMD2.G. The lentivirus containing media was collected 72 h after transfection with following filtration through 0.45 µm PVDF membrane filter (Merck, Darmstadt, Germany).

### 2.7. Lentiviral Transduction of Mammalian Cells

Transduction of T1-IM-R cells and its Cas9-expressing sublines was performed in 6-well plates with addition of 100 µL of lentivirus containing medium for each well. Cell culture medium was replaced 48 h after transduction with a fresh one containing selective antibiotics, 1 µg/mL of puromycin (fop pCW-Cas9 transduction) or 5 µg/mL of blasticidin S (for pLenti-SG-1/sgRNA transduction).

### 2.8. Induction of Doxycycline-Dependent Cas9 Expression

Doxycycline-dependent Cas9 expression in T1-IM-R/Cas9 cells was induced by adding doxycycline (Dox) to culture medium up to 1 µg/mL. Cells were cultivated in the presence of Dox for 6 days with culture medium refreshed every 2–3 days. Analysis of Cas9 expression was performed by Western Blotting.

### 2.9. Single-Cell Cloning

Single-cell clonal selection after lentiviral transduction was performed using serial dilution technique in 96-well plates as described [20]. At least four 96-well plates were used for each transduced cell line. Cells transduced with pLenti-SG1-sgRNA lentiviruses were cultivated in the presence of Dox for 6 days prior to the single-cell cloning.

### 2.10. Design of sgRNA Oligos

Sequences of sgRNA oligos for generation of CRISPR/Cas9-mediated knockdown of *FGF-2*, *FGFR1*, and *FGFR2* were taken from Wang et al. [19], Santolla et al. [21], Wahiduzzaman et al. [22], or designed using online tools from Thermo Fisher Scientific, Genscript and Origene. At least four sgRNA oligos were selected for each gene (“forward”) and reverse complement sequences were written for each one (“reverse”). Forward and reverse sequences were supplemented with “CACC” and “AAAC” cloning sites on 5′-ends respectively. The list of the oligos is presented in the Appendix A. Complementary pairs of oligos were annealed and cloned into the pLenti-SG-1 plasmid by Esp3I restriction with subsequent transformation of XL1-Blue *E. coli* cells and plasmid DNA isolation for lentivirus production. Insertion of target sequences in the recombinant plasmids was confirmed by Sanger sequencing with U6 primer (5′-GGGCAGGAAGAGGGCCTAT-3′) (Appendix A).

### 2.11. Cellular Survival MTS-Based Assay

GIST cells indicated above were seeded in 96-well flat-bottomed plates (Corning Inc., Corning, NY, USA) and allowed to attach and grow for 24 h. The cells were then cultured for 24–48 h with indicated concentrations of the RTK inhibitors or DMSO (control). Next, MTS reagent (Promega, Madison, WI, USA) was introduced to the culture medium for at least 1 h to assess the live cell numbers. The cellular viability was assessed at 492 nm on a MultiScan FC plate reader (Thermo Fisher Scientific, Waltham, MA, USA). Resulting IC_50_ values were defined as the compound concentration required to inhibit cellular growth by 50% in 24–48 h. The data was normalized according to the DMSO-treated (control) group.

### 2.12. Real-Time Monitoring of Cell Proliferation

The growth curves of GIST cells were analyzed by using iCELLigence system (ACEA Biosciences, Inc., San Diego, CA, USA). For this, GIST cells were seeded in electronic microtiter plates (E-Plate; Roche Diagnostics, GmbH, Mannheim, Germany) for 24 h to obtain the growth baseline reading. Next, GIST cells were treated with RTKi indicated above, alone or in combination, for 48–72 h. The cells treated with DMSO served as a control. Cell index (CI) measurements were performed with a signal detection set for every 30 min until the end of the experiment (72 h). Normalized cell index (NCI) values were analyzed by RTCA Data Analysis Software version 1.0 (ACEA Biosciences, Inc., San Diego, CA, USA)).

### 2.13. Western Blotting

For Western blotting analysis, whole-cell extracts were prepared by scrapping the cells growing as monolayer into RIPA buffer (25 mM Tris-HCl pH 7.6, 150 mM NaCl, 5 mM EDTA, 1% NP-40, 1% sodium deoxycholate, 0.1% SDS), supplemented with protease and phosphatase inhibitors. The cellular lysates were incubated for 1 h at 4 °C and then clarified by centrifugation for 30 min at 13,000 rpm at 4 °C. Protein concentrations were measured by the Bradford assay. The samples containing 30 μg of protein were resolved on 4 to 12% Bis-Tris or 3 to 8% Tris-acetate NuPAGE gels (Invitrogen, Carlsbad, CA, USA), transferred to a nitrocellulose membrane (Bio-Rad, Hercules, CA, USA), probed with specific antibody, and visualized by enhanced chemiluminescence (Western Lightning Plus-ECL reagent, Perkin Elmer, Waltham, MA, USA). Densitometric analysis of Western blotting images was performed by using the NIH ImageJ software (version 1.49) (Bethesda, MD, USA).

### 2.14. Immunofluorescence Staining

Cells were seeded on glass coverslips coated with poly-L-lysine (Sigma-Aldrich, St. Louis, MO, USA) and allowed to attach for 48 h before treatment. After washing with PBS, cells were fixed in 2% paraformaldehyde in PBS for 10 min at room temperature, washed with PBS and additionally fixed with ice-cold methanol for 10 min. Methanol was removed with 5 × 5 min PBS washing. Alternatively, washed cells were fixed in 4% paraformaldehyde in PBS for 30 min at 4 °C and further permeabilized with 0.5% Triton X-100 for 5 min. After 30 min of blocking with 10% normal goat serum in PBS, cells were washed and incubated with primary antibodies overnight at 4 °C. Next day the cells were washed with PBS, incubated with Alexa Fluor 488, or TexRed-conjugated secondary antibodies (Invitrogen, Carlsbad, CA, USA) for 30 min at room temperature in the dark. After brief DAPI (Sigma-Aldrich, St. Louis, MO, USA) staining, the coverslips were mounted on glass slides and cells visualized on an Olympus BX63 fluorescence microscope. Images were captured using a Spot advanced imaging system or Nikon N-SIM confocal system.

### 2.15. Colony Formation Assay

The cells were pretreated with IM alone and in the presence of anti-FGF-2 or BGJ398 for 72 h as shown earlier. The cells were trypsinized, washed with PBS twice and seeded for colony formation (approximately 500 cells for p100 culture dish) in complete DMEM-Hi medium with 10% FBS (Thermo Fisher Scientific, Waltham, MA, USA). The colonies were fixed by 70% ethanol, stained with Giemsa stain and counted using Colony V1.1 software (Fujifilm, Tokyo, Japan) after 10 days of incubation. The experiments were performed in triplicates.

### 2.16. RNA Extraction and Real-Time Quantitative PCR

Total RNA was extracted either from GIST T-1 and GIST T-1R cells and converted into cDNA as previously described [13]. A quantity of 1 µL template cDNA was used in a real-time qPCR reaction with 5x qPCRmix-HS SYBR (PB025, Evrogen, Moscow, Russia) and 10 mM each forward and reverse primers for experimental or control genes. Real-time qPCR was carried out using the CFX96 Real-Time detection system (Bio-Rad, Hercules, CA, USA), according to the manufacturer’s protocol. The absolute levels of each mRNA were normalized relative to GAPDH. Generation of quantitative data was based on the number of cycles needed for the amplification-generated fluorescence to reach a specific threshold of detection (the Ct value).

### 2.17. RNA-Seq Library Preparation, Sequencing and Bioinformatics Pipeline

Cells were collected by centrifugation and twice washed with PBS and immediately deep-frozen by liquid nitrogen. Cell pellets were used for total RNA isolation by means of RNeasy Mini Kit (Qiagen, Hilden, Germany) according to the manufacturer’s instructions.

The quality of total RNA was evaluated using the Bioanalyzer 2100 (Agilent, Santa Clara, CA, USA). The quantity and purity of RNA was estimated on a NanoPhotometer (Implen), and 800 ng of RIN ≥7 of total RNA was used for library construction using NEBNext^®^ Poly(A) mRNA Magnetic Isolation Module and NEBNext^®^ Ultra II™ Directional RNA Library Prep Kit for Illumina (New England Biolabs, Ipswich, MA, USA) according to the manufacturer’s instruction. The quality of libraries was verified using the Bioanalyzer 2100 (Agilent, Santa Clara, CA, USA) and the yield was validated by qPCR. Libraries were then sequenced on NovaSeq6000 (Illumina, San Diego, CA, USA) with pair-end 61 bp reading.

Quality of sequenced reads was accessed through FastQC v0.11.5. Unwanted adapter and rRNA matching reads were filtered out with Trimmomatic = 0.38. Trimmed reads were aligned to human genome assembly hg38 (UCSC primary chromosomes) with Hisat2 v2.1.0 and counted with HTSeq v2.0.1 against Gencode v43 basic annotation. Further analyses, including FPKM, TMM normalization, differential expression, and functional enrichment, were conducted in R environment (packages edgeR v3.42.2 and clusterProfiler v4.8.1). We consider the significance threshold for differentially expressed genes as FDR < 0.05 and fold change > 2, unless specific parameters are specified. Raw and processed transcriptome data is accessible through GEO NCBI under id GSE247170.

### 2.18. High-Throughput DNA Sequencing

Deep sequencing was performed using the Ion Torrent platform (ThermoFisher, Waltham, MA, USA) following established protocol [23]. The protocol includes preparation of libraries of genomic DNA fragments, clonal emulsion PCR, sequencing, and bioinformatics analysis of the obtained results. Target DNA fragment libraries were prepared using Ion Ampliseq ultra-multiplex PCR technology with Ion AmpliSeq Comprehensive Cancer Panel (Thermo Fisher Scientific, cat. no 4477685, Waltham, MA, USA), which provides a highly multiplexed selection of 409 genes implicated in cancer research. Multiplex PCR and subsequent stages of the fragment library preparation were performed using an Ion AmpliSeq Library Kit 2.0 (Thermo Fisher, Waltham, MA, USA), according to the manufacturer’s protocol. Aliquots from the prepared libraries were subjected to clonal amplification on microspheres in the emulsion on the Ion Chef Instrument (ThermoFisher, Waltham, MA, USA). Sequencing was performed on the Ion S5 genomic sequencer according to the manufacturer’s protocol (Thermo Fisher, Waltham, MA, USA) with average base coverage depth of >850×. The results were analyzed with Torrent Suite software (version 5.10.1) consisting of Base Caller (the primary analysis of the sequencing results); Torrent Mapping Alignment Program—TMAP (alignment of the sequences to the reference genome GRCh37/hg19); and Torrent Variant Caller (analysis of variations in nucleotide sequences), with the cut-off for variant allele frequency set at 0.1, and minimum read depth of the variant allele set at 5. Genetic variants were annotated with ANNOVAR software (2019 Oct24 release) [24]. Visual data analysis, manual filtering of sequencing artifacts, and sequence alignment were performed using the Integrative Genomics Viewer (IGV) [25].

### 2.19. GIST Xenograft Models

Subcutaneous human tumor xenografts were generated via s.c. inoculation in the flank areas of 5–8-week-old female nu/nu mice with 100 μL of 1 × 10^7^ cells/mL GIST cell suspension in Dulbecco’s phosphate buffered saline. The protocols were approved by the Committee for Ethics of Animal Experimentation, and the experiments were conducted in accordance with the Guidelines for Animal Experiments in N.N. Blokhin National Medical Research Center of Oncology (Ethical Committee protocol 05p-17/05/2023). All s.c. tumors were allowed to reach a volume of ~200 mm^3^ before the randomization of mice into treatment groups. Mice were orally administered either 50 μL of vehicle (negative control), IM (50 mg/kg), BG J398 (20 mg/kg), or a combination of the drugs indicated above. The animals were randomized into groups of 10 animals for each treatment regimen. The tumor volume, weight, and general health of the animals were recorded. After the mice had been sacrificed, tumors were excised and subjected to a histopathologic examination. Formalin-fixed, paraffin-embedded (FFPE) tissues were sectioned at 4 μM for hematoxylin and eosin (H&E) staining and IHC-staining for cleaved caspase-3. The images were captured using Aperio’s ScanScope XT (Vista, CA, USA).

### 2.20. Statistics

All the experiments were repeated a minimum of three times. The results are presented as the mean ± standard error (SE) for each group. Differences were considered significant at *p* < 0.05.

## 3. Results

### 3.1. Imatinib(IM)-Resistant GIST Cells Lacking Secondary KIT Mutations Acquire Sensitivity to pan-FGFR Inhibitors after a Long-Term Culture with IM

We previously showed the activation of FGF-signaling pathway in GIST as an alternative mechanism underlying secondary resistance to IM in GISTs lacking secondary *KIT* mutations. This was evidenced both in vitro and in vivo, thereby illustrating a rationale for using FGFR inhibitors (FGFRi) to restore GIST sensitivity to IM [13,14,15]. Of note, IM-resistant GIST T-1 cells (further named as GIST-R1) were non-sensitive to inhibition of FGF-signaling in the absence of IM in culture, thereby suggesting IM-induced autocrine activation of FGF-signaling pathway in this particular GIST subline. However, further culturing of GIST-R1 cells with IM (for over 12 months) dramatically changed their sensitivity to BGJ 398, which was evidenced by significant increase in the number of floating cells after BGJ 398 treatment (Appendix A). Similarly, expression of apoptotic markers (e.g., cleaved forms of caspase-3 and PARP) was significantly increased in BGJ 398-treated IM-resistant GISTs (Appendix A). Of note, GIST-R2 cells exhibited similar morphology when compared with GIST-R1 cells (as shown in Appendix A). Importantly, we observed the reduced expression of c-Kit in ~20% of GIST-R2 cells, as shown in Appendix A, thereby indicating clonal heterogeneity of GIST cells as a result of extended culture of cancer cells with IM.

To examine these findings more precisely, we performed a single cell cloning procedure and established a clone of GIST cells that exhibited an exceptional sensitivity to the FGFR-inhibitors. This GIST clone was further named GIST-R2 due to its remaining resistance to IM. Indeed, we observed a substantial (>14,000 fold) decrease of IC50 values for BGJ 398 in GIST-R2 cells, when compared with parental GIST-R1 cells (Figure 1 and Table 1). Immunoblot analysis revealed the substantial increase in expression of apoptosis markers in BGJ 398-treated GIST-R2 cells (Figure 2A, right panel) when compared with BGJ-398-treated GIST-R1 cells (Figure 2A, left panel). As expected, IM has no cytotoxic effects on both GIST-R1 and R2 cells, thereby revealing the retained resistance to inhibition of KIT-signaling in both types of cancer cells. The pro-apoptotic activity of BGJ 398 in GIST-R2 cells correlated with the crystal violet staining and illustrated an exclusive sensitivity of GIST-R2 cells to this FGFRi (Figure 2B—left panel). As expected, BGJ 398 effectively interfered with the growth kinetics of GIST R-2 cells (Figure 2C—right panel) and has no inhibitory effect on GIST-R1′s proliferation capacity (Figure 2C—left panel). Interestingly, when IM was used in combination with BGJ 398, a significant decrease in growth kinetics of GIST-R1 cells was observed (Figure 2—left panel), thereby revealing IM-induced activation of FGFR-signaling pathway in IM-resistant GIST.

In order to confirm the on-target effect of FGFR inhibition in GIST-R2 cells, we examined the signaling pathways responses upon BGJ 398 treatment. In contrast to GIST-R1, GIST-R2 cells exhibited a decreased expression of the phosphorylated forms of AKT, MAPK, STAT1, and S6 ribosomal protein after BGJ 398 treatment (Figure 3).

In agreement with our data illustrating a substantial increase in sensitivity of GIST-R2 cells to BGJ 398, we also observed similar effects for the other FGFRi targeting all 4 types of FGFR receptors. Indeed, treatment of GIST-R2 cells with AZD 4547 and Futibatinib (TAS-120) inhibited their survival (Figure 4A–B, Appendix A) and growth kinetics (Figure 5), thereby suggesting about the aberrant activation of FGF-signaling pathway. As expected, parental IM-resistant GISTs were non-sensitive to selective or pan-FGFR inhibitors, as well (Figure 4A–B, Appendix A), thereby revealing the acquisition of sensitivity to pan-FGFR-inhibitors after the prolonged culture of IM-resistant GIST with IM. Of note, selective FGFR1 or FGFR4 inhibitors (e.g., PD 173074 and H3B-6527, respectively) did not have a significant impact of the survival and proliferative activity of GIST-R2 cells (Figure 4A–B and Figure 5, Appendix A), thereby suggesting the aberrant activation of the FGF-signaling pathway in GIST-R2 cells via the FGF-2/FGFR2-signaling axis.

Altogether, this data illustrates that extended (over 6 months) culture of IM-resistant GISTs with IM induces clonal heterogeneity of cancer cells by generating a clone exhibiting an aberrant activation of the FGFR-signaling pathway, thereby rendering these cells extremely sensitive to pan-FGFR inhibitors.

### 3.2. Molecular Profiling of IM-Resistant GIST Cells Sensitive to FGFR Inhibitors

To identify the molecular mechanisms involved in the regulation of FGFR signaling in GIST-R2 cells rendering them sensitive to pan-FGFR inhibitors, we initially performed next generation sequencing to examine whether these cells acquired activatory *FGFR* mutations. The mutations in *KIT* gene were also examined in both IM-resistant GIST cell sublines. As expected, the characteristic deletion in the region inside exon 11 (V560-Y578) was found in both IM-resistant GIST sublines, whereas the well-known FGFR mutations affecting GIST sensitivity to the FGFR inhibitors were not detected in either GIST-R1 or GIST-R2 cells. As expected, similarly to parental GIST T-1 cells, both GIST-R1 and -R2 cells exhibited homozygous deletion of *TP53* (besides the last two codons).

Next, we performed the transcriptome profiling of naive (i.e., IM-sensitive) GIST-T1 cells, and their IM-resistant counterparts (GIST R-1 and GIST-R2 cells). Indeed, GIST-R2 cells also exhibited the most profound differences in the transcripts when compared with GIST-R1 cells and IM-naive GIST T-1 cells (Figure 6A). When analyzing the FGF/FGFR transcripts, we found that overexpression of FGF1 ligand in GIST-R2 cells was the most significant change when compared with GIST-R1 and naive GIST T-1 cells (Appendix A). Moreover, the comparative analysis of the whole transcriptome in GIST-R1 and R-2 cells allowed determination of the transcripts overexpressed in GIST-R2 cells when compared with GIST-R1 cells (Figure 6B). Based on the literature data (as discussed below), some of these hits might be involved in regulation of FGFR signaling in GIST cells. In particular, this belongs to the alpha-2-macroglobulin (A2M), protein 1 containing signal peptide-cub-EGF-like domain (SCUBE1), SIX2, etc. The regulatory role of these proteins in FGFR-signaling in GISTs is discussed below. Importantly, the major differences between GIST R-1 and R-2 cells were observed for the transcripts that belong to the PI3K/Akt- and MAPK-signaling pathways (Figure 6C). The detailed analysis of these changes for both of the aforementioned signaling pathways is shown in Appendix A. The significant changes in the transcripts that belong to these pathways are shown in Figure 7. Of note, FGF1 transcript was on the top of 2 genes up-regulated in GIST-R2 cells, thereby suggesting that activation of PI3K/Akt- and MAPK-signaling cascades is ongoing via the FGF-signaling axis (Figure 7A,B). As expected, in GIST-R2 cells FGF1 transcript was found on the top one of the up-regulated transcripts that belong to the FGF-signaling pathway (Figure 7C). The other ligands of the FGFR-signaling pathway (e.g., FGF-4, -6 and -23) were also up-regulated in GIST-R2 cells when compared with GIST-R1 cells, thereby illustrating the activation of FGFR-signaling in this particular GIST subclone.

Western blot analysis revealed that the extended culture of GIST cells with IM resulted in the gradual decrease of expression of the total and phosphorylated forms of c-Kit. This was found for GIST-R1 cells when compared with parental (IM-naive) GIST-T1 cells. Moreover, GIST-R2 exhibited profound decrease of KIT expression and activation when compared with their “precursors” (e.g., GIST-R1 cells) (Figure 8A). Altogether, this data illustrates that acquisition of IM resistance in these cells was associated with “Kit loss”. Conversely, expression of the total and phosphorylated forms of FGFR1/2 gradually increased in IM-resistant GIST cells (e.g., GIST-R1 and its subclone GIST-R2) (Figure 8A). The expression of FGFR3 and 4 did not altered in IM-resistant GIST when compared with the parental (e.g., IM-naive) GIST T-1 cells (Figure 8A), thereby reveling that activation of FGFR-signaling in IM-resistant GIST is ongoing via FGFR1/2 signaling axis. Thus, an increased sensitivity of GIST-R2 cells to pan-FGFR inhibitors might be due to hyperactivation of FGFR-signaling. This was in agreement with our findings, illustrating overexpression of FGF-2 and FGFR1/2 in GIST-R2 cells shown by Western blotting and the quantitative RT-PCR (Figure 8B–D). Moreover, overproduction of FGF-2 ligand by GIST-R2 cells suggested about autocrine activation of FGF2-FGFR1/2 loop, thereby rendering these cells sensitive to FGFR-inhibitors. As expected, BGJ 398 completely abrogated FGFR-signaling cascade, which was evidenced by the absence of the phosphorylated forms of FGFR1/2 and adaptor protein FRS-2 in BGJ 398-treated GIST-R2 cells (Figure 8B).

### 3.3. Targeting of FGFR Signaling Effectively Inhibits Growth of GIST-R2 Xenografts In Vivo

Next, we examined IM-resistant GIST’s sensitivity to the FGFR-inhibitors in vivo. For this purpose, GIST-R1 and R2 cells were injected into the flanks of female adult athymic nude mice and tumors were allowed to grow for at least for 2 weeks. After 2 weeks, the animals were randomized into 4 groups, treated twice a week with a vehicle (control), or BGJ 398 (20 mg/kg) for 14 days. As expected, vehicle-treated mice (control) demonstrated a continuous increase in tumor size from the baseline over the 2-week period. In BGJ 398-treated animals, GIST-R1 tumors showed a minor regression in size when compared to the baseline, thus revealing the resistance of GIST cells to FGFR-inhibitor. Strikingly, a substantial decrease in tumor size, volume and weight were observed in BGJ 398-treated mice bearing GIST-R2 xenografts (Figure 9). In concordance with these findings, BGJ 398-treated GIST-R2 xenografts exhibited the substantial increase of areas of central necrosis as assessed by HE staining (Figure 10—left panel), whereas no evidence of central necrosis was observed in control (solvent-treated) GIST-R2 xenografts. Of note, BGJ 398-treated GIST-R1 xenografts lacked the necrotic areas, thereby revealing a selective sensitivity to BGJ 398 belongs to GIST-R2 cells. As expected, apoptosis was found as a major mechanism of cancer cell death in BGJ 398-treated GIST-R2 xenografts, which was evidenced by the immunohistochemical (IHC)-staining illustrating the substantial increase in the number of caspase-3-positive cells (Figure 10—right panel).

### 3.4. Generating of Monoclonal GIST-R2 Cells with CRISPR/Cas9-Mediated Knockout of the FGF-2, FGFR1 and FGFR2 Genes

Given that FGFR mutations were not detected in GIST-R2 cells, and taking into account an increased expression of FGFR1/2 and FGF-2 in these cells, we further generated GIST clones with CRISPR/Cas9-mediated knockout of FGF-2 and 2 types of FGFRs. This was aimed to examine the precise molecular mechanisms underlying sensitivity of GIST-R2 cells to FGFR-inhibition. For this purpose, GIST-R2 cells were transduced by doxycycline-dependent Cas9-containing lentiviruses produced in HEK293T cells by using pCW-Cas9 transfer plasmid and psPAX2 and pMD2.G helper plasmids as described in Section 2. After the puromycin selection transduced cells were used to obtain monoclonal cell populations expressing doxycycline-dependent Cas9. Overall, 32 monoclonal populations were gained, 12 of which expressed doxycycline-dependent endonuclease Cas9 and T1-IM-R/Cas9[p4-E9] clone was selected for further work (Appendix A).

T1-IM-R/Cas9[p4-E9] cells were further transduced by lentiviruses carrying sgRNA oligos to develop CRISPR/Cas9-mediated knockout of the *FGF-2*, *FGFR1*, and *FGFR2*, with subsequent selection in the presence of blasticidine S. To produce sgRNA coding lentiviruses corresponding sgRNA oligos against *FGF-2*, *FGFR1* and *FGFR2* genes were cloned into pLenti-SG-1 plasmid as described in Section 2. The results of Sanger sequencing proving the successful sgRNA oligos cloning are shown in Appendix A. Data processing was performed in Unipro UGENE 42.0 program software [26,27,28].

To initiate CRISPR/Cas9-mediated knockout of the studied genes the Cas9 expression was induced by adding of 1 μg/mL doxycycline in the cell culture medium for 6 days with the ongoing monoclonal population development and Western blotting detection of a level of the studied genes expression products. As shown in Figure 11A, we observed a significant decrease of FGFR1, FGFR2 and FGF-2 expression in the corresponding clones of GIST-R2 cells, thereby revealing the knockout of FGFR 1 and 2 and their ligand FGF-2. The knockout efficacy was also revealed by the immunofluorescence staining (Figure 11B) and real-time quantitative PCR (Figure 11C), which allowed the use of these clones for the following experiments to examine molecular mechanisms of GIST sensitivity to FGFR inhibitors. Of note, a knockout of FGFR1, 2 or FGF-2 in GIST-R2 cells did not affect their abilities to grow up in the colonies (Appendix A), invasion and migration, as well, thereby illustrating a similar phenotype of FGF-2−/−, FGFR1−/−, and FGFR2−/− GIST cells.

### 3.5. FGFR2-Mediated Signaling Is Critical for Survival, Proliferation, and Colony Formation of GIST-R2 Cells

We initially examined whether the knockout of *FGFR1*, *2* or *FGF-2* changed GIST-R2′s sensitivity to BGJ 398. We found that knockout of *FGF-2* and *FGFR1* in GIST-R2 cells did not have a significant impact on the cellular viability after BGJ 398 treatment—i.e., GIST-R2 cells remain sensitive to the FGFR inhibition, which was evidenced by crystal violet staining (Figure 12A), assessment of proliferative activity (Figure 12B) and expression of caspase-3, the well-known apoptotic marker (Figure 12C). In comparison to FGF-2−/− and FGFR1−/− cells, FGFR2−/− cells exhibited an increased survival after BGJ 398 treatment and minor decrease of the proliferative activity (Figure 12), thereby illustrating the activation of FGF-signaling via FGFR2-mediated axis and rendering these cells highly sensitive to BGJ 398. In concordance with these findings, we also observed a significant increase of IC50 values for BGJ 398 in FGFR2−/− GIST-R2 cells, when compared with wild-type or FGF-2−/− and FGFR1−/− cells, as was shown in the Appendix A.

Similarly, knockout of *FGFR2* did not have a significant impact on the anti-proliferative activity of BGJ 398, whereas the FGFR1−/− and FGF-2−/− cells exhibited a substantial decrease of proliferative activity after BGJ 398 treatment (Figure 12C). As expected, IM treatment of cell sublines with the knockout of *FGFR1*, *FGFR2* or *FGF-2* did not decrease their proliferative activity when compared with the non-treated cells, thereby revealing their resistance to IM.

To further corroborate these findings, we examined whether knockout of *FGFR1/2* and *FGF-2* have the similar impact of GIST-R2′s sensitivity to FGFR inhibition in vivo. For this purpose, GIST R-2 cells with knockout of *FGFR1*, *2* or *FGF-2* were injected into the flanks of female adult athymic nude mice and tumors were allowed to grow for at least for 2 weeks before BGJ 398 treatment. Similar to in vitro data BGJ 398 exhibited the moderate anti-tumor effect on GIST-R2 xenografts with the knockout of *FGFR2* (Figure 13), whereas BGJ 398 treatment induced a substantial decrease of tumor size in animals bearing GIST-R2 xenografts with the knockout of *FGFR1* or *FGF-2*, thereby revealing the autocrine-based activation of FGF-2/FGFR1 signaling axis as a predominant pathway of an acquired sensitivity of GIST-2 cells to BGJ 398.

Collectively, our data illustrates the aberrant activation of FGF-signaling in IM-resistant GIST after the long-term exposure to IM. This is achieved via autocrine-based activation of FGF-2/FGFR1-signaling axis and leads to significant increase of sensitivity of IM-resistant GIST to pan-FGFR inhibitors. Thus, we concluded that an acquired sensitivity of GIST-R2 cells to FGFR inhibitors in vitro and in vivo is going via FGF-2/FGFR1 signaling axis.

## 4. Discussion

Activation of FGF/FGFR signaling pathways is well-documented for a broad spectrum of human malignancies and plays an important role in carcinogenesis, tumor development, and progression [29,30,31,32,33]. Our previous studies demonstrated that an acquired resistance of GIST to IM lacking secondary *KIT* mutations might be due to the activation of FGFR-signaling pathway [13]. Li F., with co-authors observed overexpression of FGF-2 and FGFR-1 in primary GIST [16], whereas Javidi-Sharifi N. with co-authors demonstrated an increased expression of FGF-2 in IM-resistant GIST cells and also observed the synergy between KIT- and FGFR-3 inhibitors against GIST, thereby suggesting about the cross-talk ongoing between KIT and FGFR-3 [17]. In consistency with these findings, we observed that FGFR-inhibitors (e.g., BGJ 398) effectively restored GIST’s sensitivity to IM both in vitro and an in vivo, thereby providing a rationale of the combined therapies by the aforementioned receptor tyrosine kinase inhibitors (RTKi) for the selected group of GIST patients exhibiting activation of FGFR-signaling pathway and progressing during the first-line (i.e., IM-based) targeted therapy [14]. In contrast, Schöffski P. with colleagues demonstrated recently that despite of BGJ 398 alone exhibited anti-tumor activity against mouse xenograft GIST models and led to the tumor volume stabilization, no additive effects of the combination of BGJ 398 with IM were found when compared with IM treatment alone [34]. This was shown on patient-derived and GIST48 cell line-derived xenografts, as well. This might be partially explained by use of IM-sensitive GIST cells harboring specific *KIT* mutations (e.g., GIST48 cell line—*p.V560D*; *D820A*; patient-derived GIST—p.A502_Y503dup) and thereby rendering them sensitive to this c-KIT-inhibitor, which was evidenced by the decreased expression of phosphorylated KIT in these particular GIST cells after IM treatment.

Lastly, we also found that IM induced the autocrine-based activation of FGFR-signaling in GIST via overproduction and secretion of FGF-2 ligand. This was evidenced for GIST cell lines treated with IM in vitro, GIST xenografts and clinical samples, as well [15]. Of note, neutralization of FGF-2 abolished GIST resistance to IM and effectively abrogated the migratory and invasive capacities of cancer cells in vitro [15]. Thus, we concluded that IM induces autocrine-based activation of FGF-signaling in GIST and this might have a significant impact on the malignant behavior of IM-resistant GISTs and disease progression.

The intratumoral heterogeneity is currently considered as a major propeller of drug resistance because of the clonal evolution and the consequent adaptive mechanisms raising up in cancer cells in therapy-challenged tumors. Indeed, this was evidenced for a broad spectrum of human malignancies, including glioblastoma [35], colorectal cancer [36], melanoma [37], chronic myeloid leukemia [38], etc. In concordance with these findings, we report here that IM-induced continuous inhibition of KIT signaling in IM-resistant GIST-T1 cells lacking secondary *KIT* mutations induced a clonal heterogeneity of cancer cells, resulted in the appearance and future expansion of GIST subclone exhibiting “KIT loss” and activation of FGF-signaling pathway. Molecular characterization of these cells illustrated the overexpression of FGFR1/2 and its ligand FGF-2, as shown in the Figure 8, whereas the activation mutations in *FGFR1-4* were not detected in GIST-R2 cells. Activation of FGF-signaling in this particular GIST subclone was revealed by increased expression of total and phosphorylated forms of FGFR1/2 and FRS-2, a well-known adaptor protein of FGF-signaling pathway (Figure 8). Thus, despite of the absence of c-KIT signaling due to the “KIT loss” (Figure 8), GIST-R2 cells maintained the activation of PI3K/Akt- and MAPK-signaling cascades, which was evidenced on the protein and mRNA levels (Appendix A). Important, these molecular events rendered GIST cells extremely sensitive to pan-FGFR-inhibitors used alone. The strong anti-proliferative and pro-apoptotic effects were observed for all 3 pan-FGFR-inhibitors used in present study (e.g., BGJ 398, AZD 4547 and TAS-120), whereas selective FGFR1 and4 inhibitors failed to kill these cells in vitro (Figure 4 and Figure 5, Appendix A). Strikingly, we also observed potent anti-tumor effect of BGJ 398 against xenograft tumors originated from this subclone, which resulted in a substantial (>80%) decrease in tumor size and increased intratumoral apoptosis as detected by immunohistochemical staining for cleaved caspase-3 (Figure 9 and Figure 10).

To examine the molecular mechanisms of FGFR-activation in these GIST cells more precisely, we performed CRISPR/Cas9-mediated knockout of the *FGF-2*, *FGFR1* and *FGFR2*, as shown in Figure 11. The efficacy of the knockout of the aforementioned genes was revealed by western blotting, immunofluorescence staining and real-time PCR, as well (Figure 10). We found that *FGFR2*−/− GIST cells were much less sensitive to BGJ 398, when compared with *FGF2*−/− or *FGFR1*−/− cells. This was evidenced by cell viability and proliferative assays, as shown in Figure 12A,B, respectively. Similarly, we observed no evidence of apoptotic markers in FGFR2−/− GIST cells treated with BGJ 398, as shown in Figure 12C. Lastly, we found the significant increase of IC50 values for *FGFR2*−/− GIST cells treated with BGJ 398, when compared with GIST-R2 cells (Appendix A). Lastly, we revealed this finding in vivo by illustrating the lack of changes in tumor sizes of FGFR2−/− xenografts in mice treated orally with pan-FGFR inhibitor, BGJ 398 (Figure 13).

In addition to overexpression of the FGFR1/2 in GIST-R2 cells, the transcriptome analysis illustrated the up-regulation of several mRNA transcripts that might be involved in regulation of FGFR-signaling in GIST. In particular, we observed the upregulation of Signal peptide-CUB-EGF-like repeat-containing protein 1 (i.e., SCUBE1) in GIST-R2 cells (Figure 6B). SCUBE proteins are expressed as peripheral membrane proteins, where they can act as the co-receptors in promoting of the activity of the numerous growth factors mediated, in particular, by receptor tyrosine kinases, including FGFR [39], VEGFR [40]. Thus, upregulation of SCUBE1 observed in GIST-R2 cells might also amplify the FGF-signaling, thereby resulting in activation of downstream PI3K/AKT- and MAPK-signaling cascades, as was shown in Figure 6C. The upregulation of *ARHGAP18* observed in GIST-R2 cells might be result of FGF-2-mediated activation due to the well-known ability of this ligand to modulate expression of the genes that belong to Rho-family of GTPases [41]. Given that ARHGAP18 is a well-known GTPase-activating protein for RhoA and taking into account that overexpression of ARHGAP18 resulted in the promotion of cell migration [42], FGF-2-induced upregulation of ARHGAP18 in GIST-R2 cells might be responsible for the aggressive phenotype of GIST-R2 cells. Analyzing proteins whose expression was increased in GIST-R2 cells, we also found several proteins that may confer negative feedback of FGF1/2-induced activation of FGFR-signaling. In particular, we observed the upregulation of alpha-2-macroglobulin (A2M) which has previously been reported to bind with FGF-2 ligand and thereby interfere with its interaction with the corresponding receptors ([43] and reviewed in detail in [44]). Given that we found previously a substantial increase of FGF-2 production and secretion in IM-treated GIST cells in vitro and in vivo [14,15], and taking into account a significant increase of FGF1 observed in GIST-R2 cells when compared with GIST-R1 and naïve GIST-T1 cells (Figure 7, Appendix A), an increase of A2M in GIST-R2 cells might represent the negative feedback suppressed activation of FGF-signaling in IM-resistant GISTs. This data is in consistency with findings of Kostas M., with colleagues who demonstrated an increased expression of A2M and heparan sulfate proteoglycan CD44 in FGF1 stimulated cells, thereby conferring negative feedback of FGF1 signaling [45].

Thus, we found that continuous inhibition of KIT-signaling pathway in GIST T-1 cells induced clonal heterogeneity of GIST which resulted in the arisement of GIST subclone with KIT loss and exhibiting activation of FGFR-pathway ongoing via FGFR1/2-signaling axis. The last one was responsible for the maintenance of PI3K/Akt- and MAPK-signaling cascades. Despite of KIT loss rendered these GIST cells resistant to IM, FGFR-overactivated cells became extremely sensitive to pan-FGFR-inhibitors both in vitro and in vivo, which in turn raising up the possibility to re-evaluate the effectiveness of pan-FGFR-inhibitors to improve the second-line therapeutic strategies for selected subgroup of GIST patients (e.g., IM-resistant GIST lacking secondary *KIT* mutations and exhibiting the activation of FGF-signaling pathway). Thus, our current data is in concordance with other studies illustrating that a long-term use of IM is raising up the clonal evolution of cancer cells to escape the cytotoxic effects of anti-cancer therapy. Indeed, besides GIST, patients with chronic myeloid leukemia who develop resistance to imatinib mesylate also frequently develop clonal evolution as a mechanism of resistance to the targeted-based therapies [46,47,48].

## 5. Conclusions

We showed here that long-term culture of GIST T-1 cells with IM induced clonal heterogeneity of cancer cells resulting in the appearance of the clone exhibiting the activation of FGFR-signaling pathway associated with KIT loss. As an outcome of these events, the clone retained resistance to IM and acquired sensitivity to pan-FGFR inhibitors both in vitro and in vivo. This illustrates a rationale to re-evaluate the effectiveness of FGFR-inhibition for the selected subgroup of GIST patients lacking secondary *KIT* mutations and exhibiting the signs of activation of FGFR-signaling as a result of the long-term inhibition of KIT-signaling pathway.

## Figures and Tables

**Figure 1 cancers-15-05354-f001:**
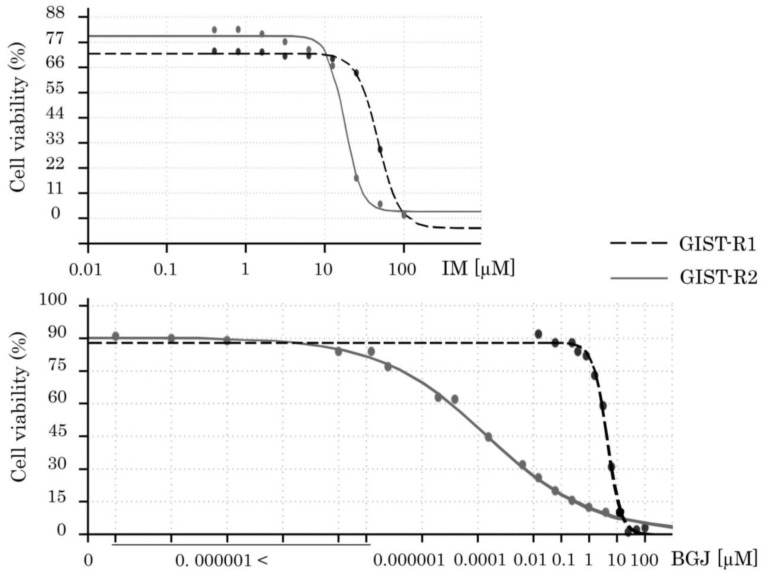
MTS-based viability assay in IM-resistant GIST-R1 vs. GIST-R2 cells. Cells were treated with the indicated concentrations of IM (upper figure) or BGJ 398 (bottom figure) and assessed after 72 h of treatment, with the data normalized to DMSO-treated controls. Values are the means ± standard deviation (n = 4).

**Figure 2 cancers-15-05354-f002:**
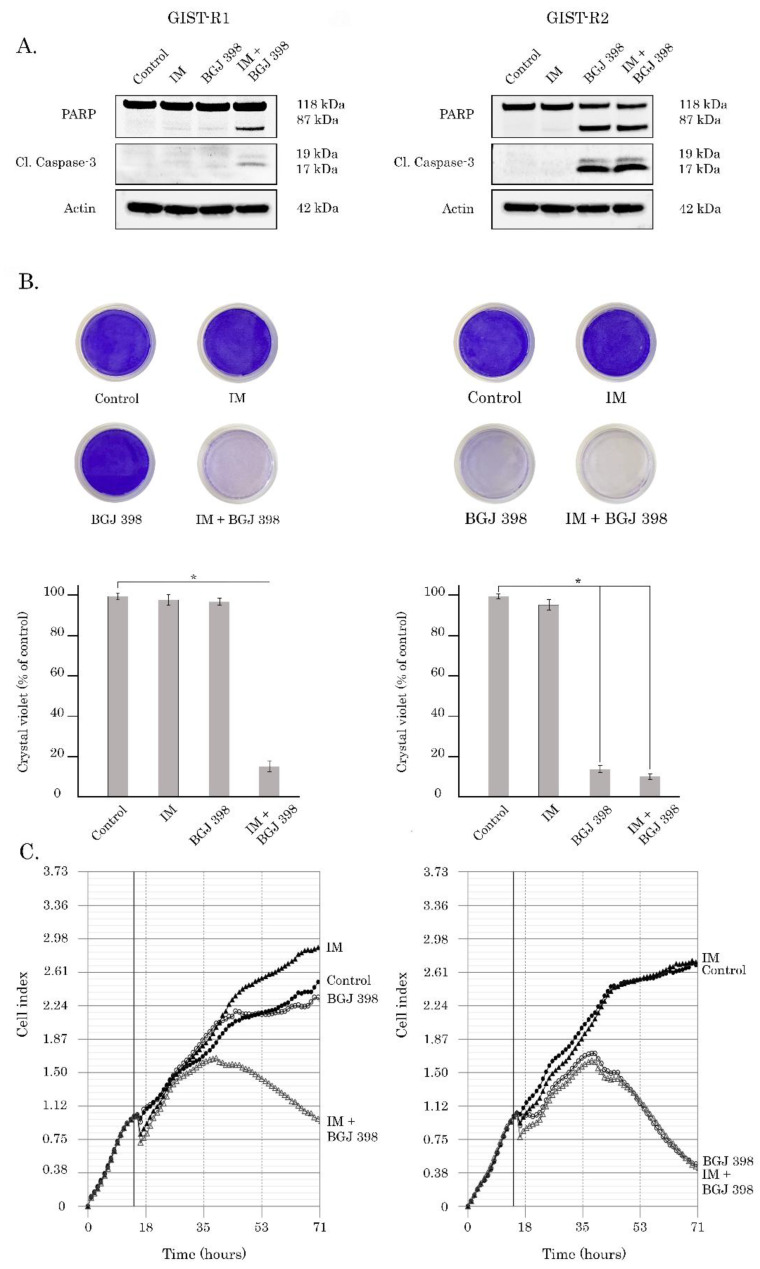
BGJ 398 induces cell death of GIST-R1 cells when used in combination with IM (left panel), whereas GIST-R2 cells were killed by BGJ 398 used alone (right panel). (**A**) Immunoblot analysis for apoptosis markers (cleaved forms of PARP and caspase-3) in GIST-R1 and R-2 cells after treatment with DMSO (control), IM, BGJ 398 alone, and in combination for 72 h. Actin stain was used as a loading control; (**B**) Upper panel—Representative images of crystal violet staining of GIST-R1 cells (left) and GIST-R2 cells (right) that were treated with IM or BGJ 398 alone or in combination for 72 h. The cells treated with DMSO were used as a control. The culture dishes were fixed with ice-cold 100% methanol, stained with crystal violet, and photographed. Lower panel—quantification of crystal violet staining of GIST cells, as shown in the upper panel. *: *p* ≤ 0.001; (**C**) Changes in growth kinetics of GIST-R1 (left) and GIST-R2 (right) cells treated with DMSO (control), IM or BGJ 398 alone and in combination. The uncropped blots are shown in Appendix A.

**Figure 3 cancers-15-05354-f003:**
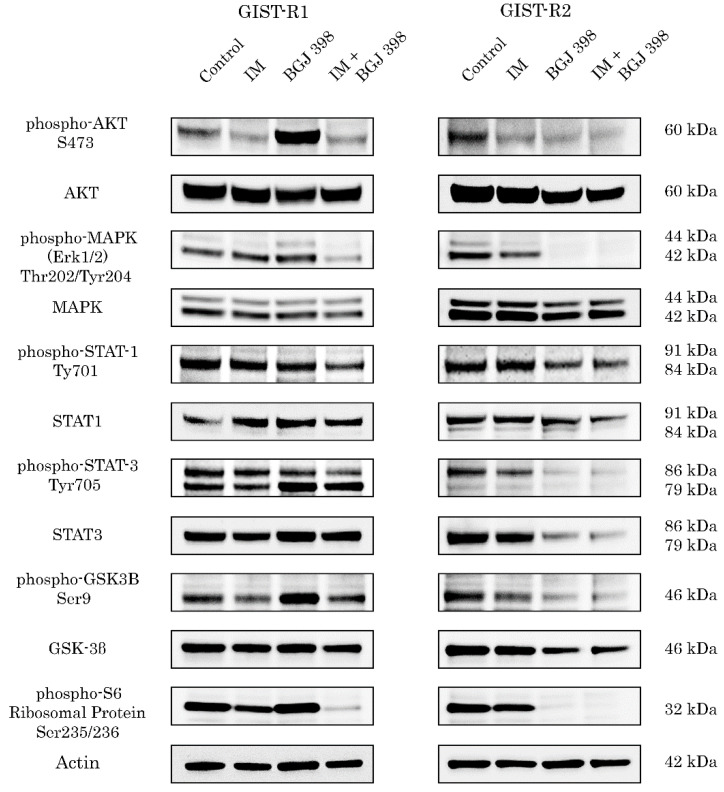
The effects of IM, BGJ 398 used alone or in combination on FGFR-signaling pathway in GIST R-1 cells (Left) and GIST-R2 cells (right). Cells were treated with aforementioned RTKi for 72 h, lysed with RIPA buffer as shown in Section 2 and processed for western blotting. The membranes were stained with corresponding primary Abs. Actin stain was used as a loading control. The uncropped blots are shown in Appendix A.

**Figure 4 cancers-15-05354-f004:**
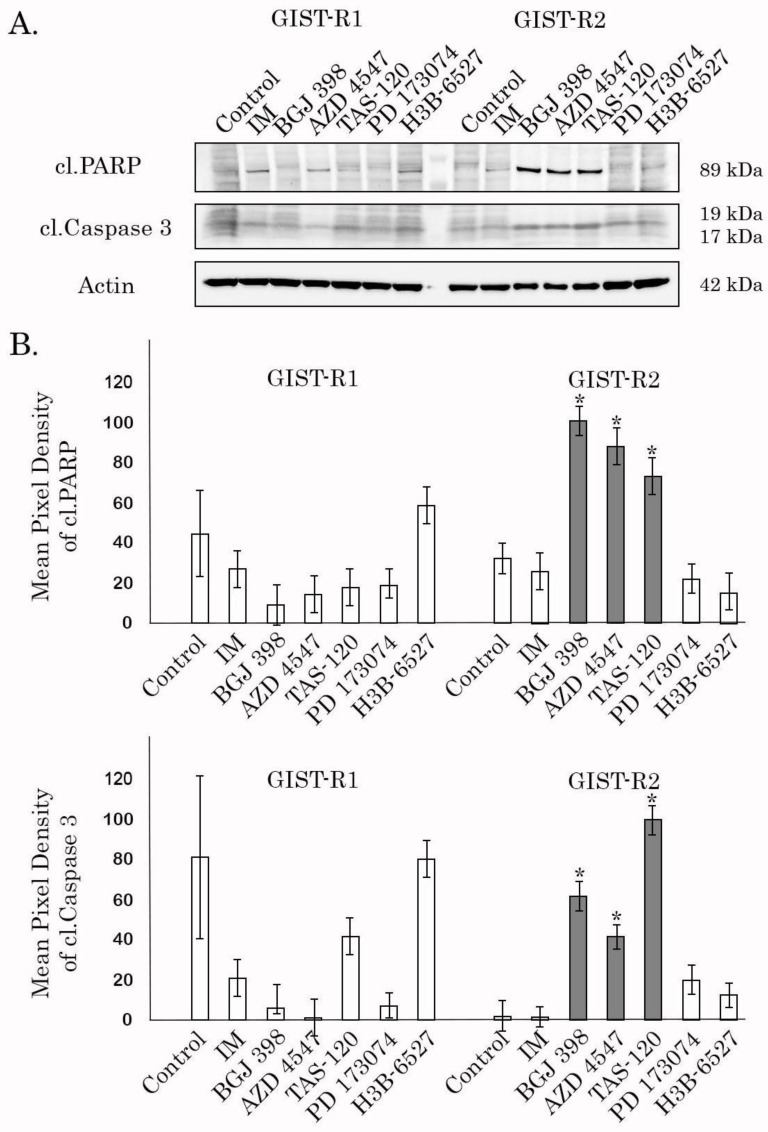
Pan-FGFR inhibitors exhibit substantial cytotoxicity against GIST-R2, but not GIST-R1 cells. (**A**) Immunoblot analysis for apoptosis markers (e.g., cleaved forms of PARP and caspase-3) in GIST-R1 and GIST-R2 cells treated with solvent (DMSO) (control), IM (1 μM) BGJ 398 (1 μM), AZD 4547 (1 μM), TAS-120 (1 μM), PD 173074 (10 μM) and H3B-6527 (1 μM) for 48 h. Actin stain is used as a loading control. (**B**) Quantification by mean pixel density of cleaved forms of PARP (upper panel) and caspase-3 (lower panel) in GIST-R1 and R2 cells as shown in Figure 1A. Values are means ± SD, n = 3. * *p* < 0.05 vs. untreated cells. The uncropped blots are shown in Appendix A.

**Figure 5 cancers-15-05354-f005:**
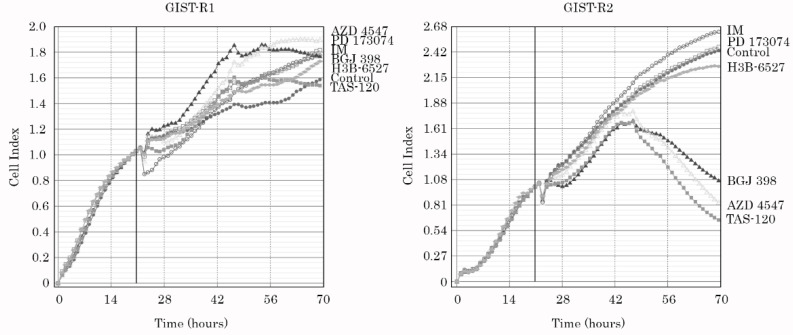
Pan-FGFR inhibitors (e.g., BGJ 398, AZD 4547 and TAS-120) exhibited potent anti-proliferative activity against GIST-R2 cells, and remained non-active against GIST-R1 cells. Changes in growth kinetics of GIST-R1 (**left**) and GIST-R2 (**right**) treated with pan- or selective FGFR-inhibitors were assessed by using the iCELLigence system (ACEA Biosciences, San Diego, CA, USA). The vertical lines represent a time-point when the RTKi were introduced into the cell cultures.

**Figure 6 cancers-15-05354-f006:**
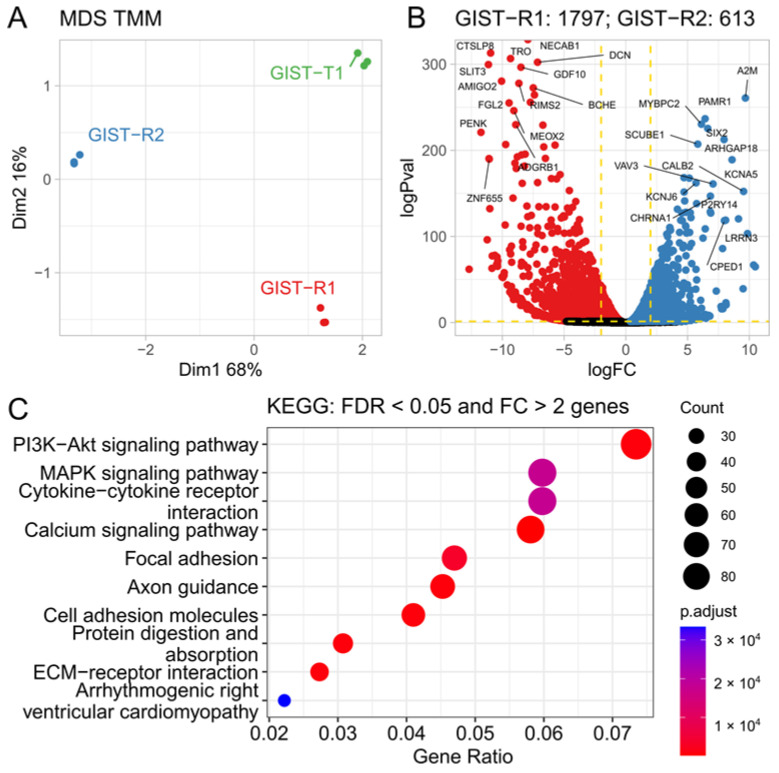
Transcriptional alterations in GIST cell lines. (**A**) MDS (multidimensional scaling) plot on TMM (trimmed means of M values) normalized counts. All three samples are clearly segregated by the first two dimensions. (**B**) Differential expression (DE) results between GIST-R1 and R2. Top DE genes are labeled. Dashed lines indicate thresholds for differentially expressed (DE) genes selection FRD < 0.05 and |logFC| > 2. Red and blue colored dots correspond to DE genes up-regulated in GIST-R1 and GIST-R2, respectively (**C**) KEGG pathway enrichment analysis for DE genes; top 10 pathways are shown. *p*-values were corrected with Benjamini–Hochberg procedure.

**Figure 7 cancers-15-05354-f007:**
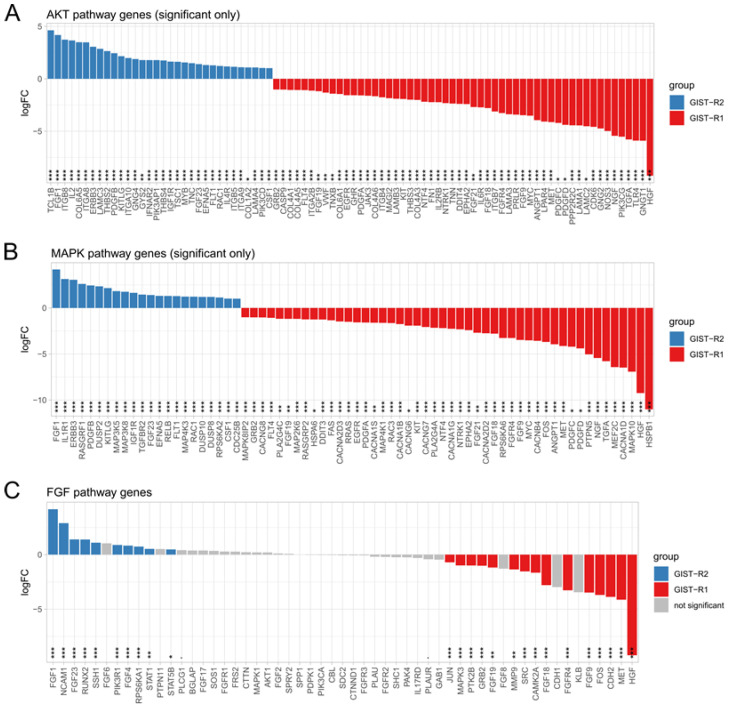
Gene expression changes in PI3K/Akt, MAPK and FGF signaling pathways. *** FDR < 0.001, ** FDR 0.001–0.01, * FDR 0.01–0.05, not significant FDR > 0.05.

**Figure 8 cancers-15-05354-f008:**
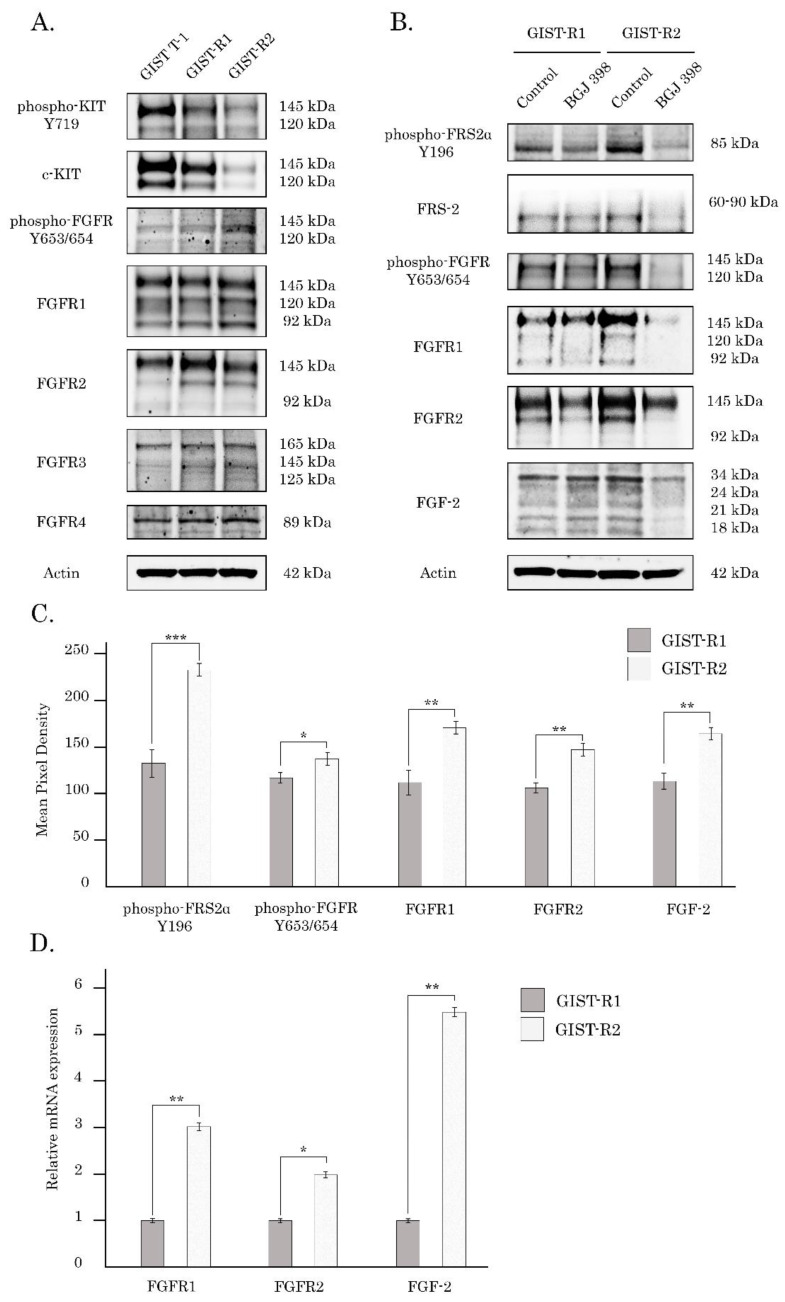
IM down-regulates KIT- and activates FGFR-signaling in IM-resistant GIST, thereby rendering cells sensitive to BGJ 398, pan-FGFR-inhibitor. (**A**) Expression of total and phosphorylated KIT was gradually decreased in GIST-R1 and R2-cells when compared with naive GIST T-1 cells, whereas expression of total and phosphorylated FGFR1/2 was gradually increased in GIST-R1 and R-2 cells. (**B**) The inhibitory effect of BGJ 398 of FGFR-signaling was much more prominent in GIST-R2 cells when compared with GIST-R1 cells (**C**) Quantification analysis of the protein expression, as shown in Figure 8B. (**D**) Changes in the relative expression level of FGFR-1, FGFR-2 and FGF-2 in GIST R-1 vs. R2-cells, as determined by RT quantitative PCR. Gene of glyceraldehyde-3-phosphate dehydrogenase (GAPDH) was amplified as an internal control. Data is expressed as the mean ± SD of three independent experiments. * *p* ≤ 0.05; ** *p* ≤ 0.01, *** *p* ≤ 0.001. The uncropped blots are shown in Appendix A.

**Figure 9 cancers-15-05354-f009:**
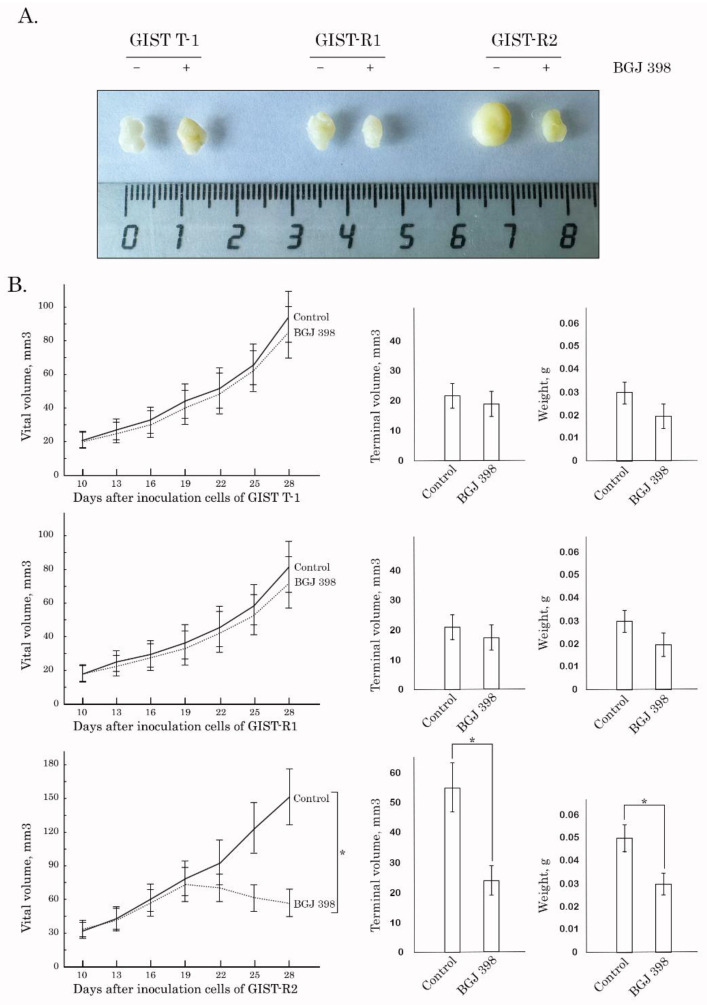
Antitumor effects of BGJ 398 in a nude mice xenograft GIST models. After subcutaneous inoculation of naive GIST T-1, GIST-R1, and GIST-R2 cells (day 14), nude mice were randomized into 6 groups (n = 4) and administered i.p. 100 μL of vehicle (negative control) or BGJ 398 (20 mg/kg). The changes in tumor sizes were calculated as a percentage of the baseline. The tumor volume in each group was assessed by calipers and calculated as length × width × width × 0.5. Results were expressed as the mean volume and weight of tumors (mean ± SE, n = 5; *: *p* < 0.001) compared to control (vehicle-treated) animals. (**A**) Representative images of the final tumor volumes in each experimental group. (**B**) Dynamics of the tumors growth, tumors volumes and weights for each experimental group.

**Figure 10 cancers-15-05354-f010:**
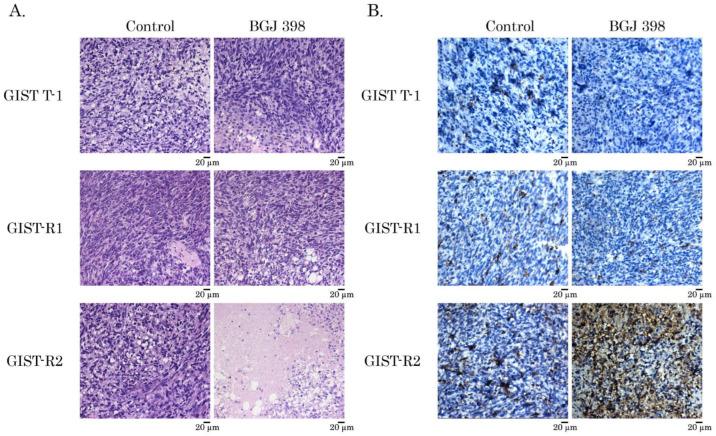
BGJ 398 exhibits potent anti-tumor and pro-apoptotic activities in IM-resistant GIST-R2 xenografts. (**A**) Representative images of hematoxylin and eosin-stained IM-naïve (GIST T-1) and resistant GIST-R1 and GIST-R2 xenografts treated for 7 days with BGJ 398, a selective pan-FGFR inhibitor (20 mg/kg). (**B**) Representative images of cleaved caspase-3 found by the IHC-staining of IM-naïve (GIST T-1) and resistant (e.g., GIST-R1 and GIST-R2) xenografts treated with BGJ 398 for 3 days.

**Figure 11 cancers-15-05354-f011:**
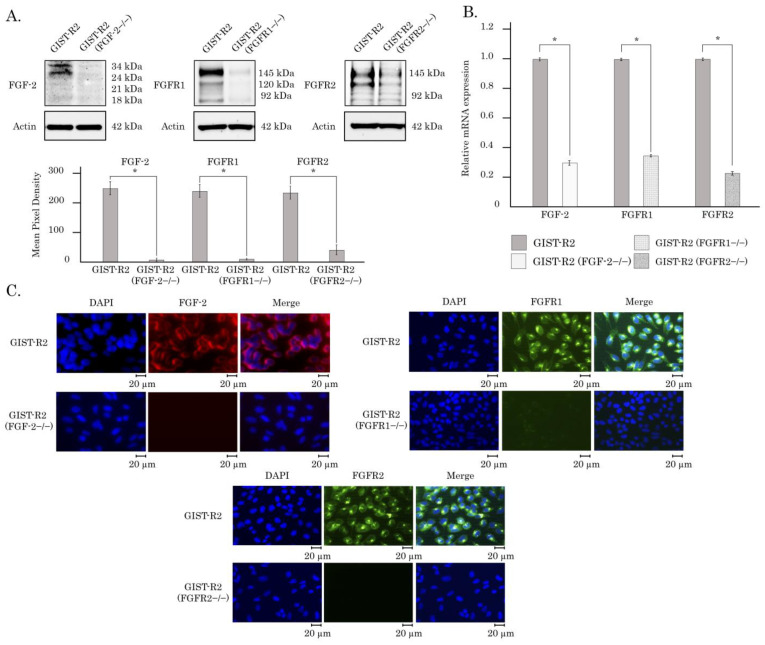
Knockout of *FGFR1*, *FGFR2* and *FGF-2* in GIST-R2 cells. (**A**) Immunoblot analysis of the total forms of FGFR1, FGFR2 and FGF-2 in GIST-R2 cells and the knock-outed cell lines. Actin stain was used as a loading control. Shows quantification of protein expression (FGFR1, FGFR2, FGF-2) based on average pixel density. * *p* < 0,001. (**B**) Changes in the expression level of *FGFR1*, *FGFR2* and *FGF2* genes in GIST-R2 cells and knockout cell lines, as determined by reverse transcription quantitative polymerase chain reaction. Actin was amplified as an internal control. Values are means ± SD, n = 3. * *p* < 0.001. (**C**) Analysis of the expression of FGFR1, FGFR2 and FGF-2 proteins in GIST-R2 cells and knockout lines by immunofluorescence microscopy. Cell nuclei were outlined by staining with DAPI. Magnification 40×, scale bars 20 μM. The uncropped blots are shown in Appendix A.

**Figure 12 cancers-15-05354-f012:**
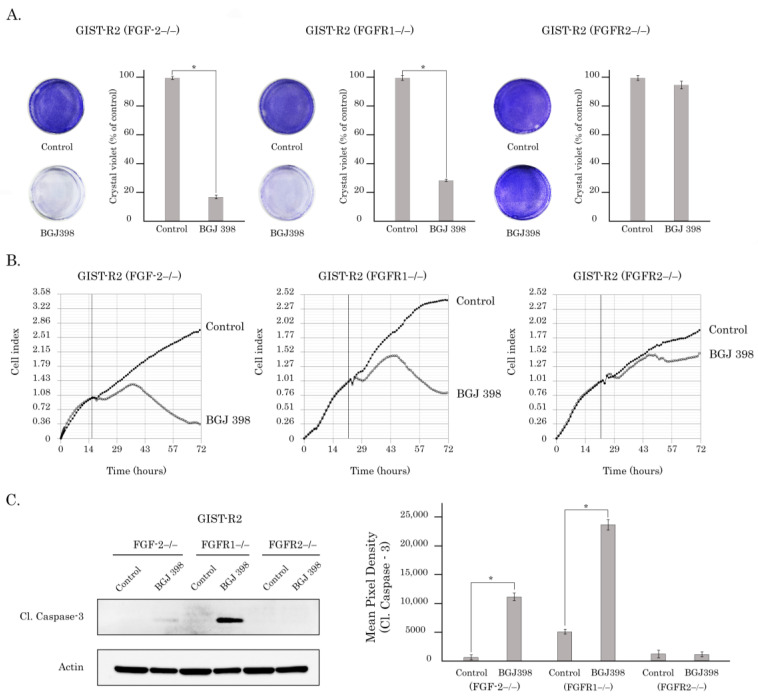
BGJ 398 provides the cytotoxic, pro-apoptotic and anti-proliferative effects in GIST-R2 cells in vitro via FGFR2-signaling axis. (**A**). Representative images of crystal violet staining of FGF-2−/−, FGFR1−/−, and FGFR2−/− cells treated with solvent DMSO (control) or BGJ 398 for 72 h and its quantification analysis. *: *p* ≤ 0.001; (**B**). Growth kinetics of FGF-2−/−, FGFR1−/−, and FGFR2−/− cells treated with solvent DMSO (control) or BGJ 398 and assessed by the iCELLigence system (ACEA Biosciences, Santa Clara, CA, USA). The vertical lines represent a time-point when BGJ 398 was introduced into the cell cultures (**C**) Left—Immunoblot analysis for cleaved caspase-3 in FGF-2−/−, FGFR1−/−, and FGFR2−/− cells after treatment with DMSO (control), or BGJ 398 for 72 h. Actin stain was used as a loading control. Right—quantification analysis of cleaved caspase-3 expression. The uncropped blots are shown in Appendix A.

**Figure 13 cancers-15-05354-f013:**
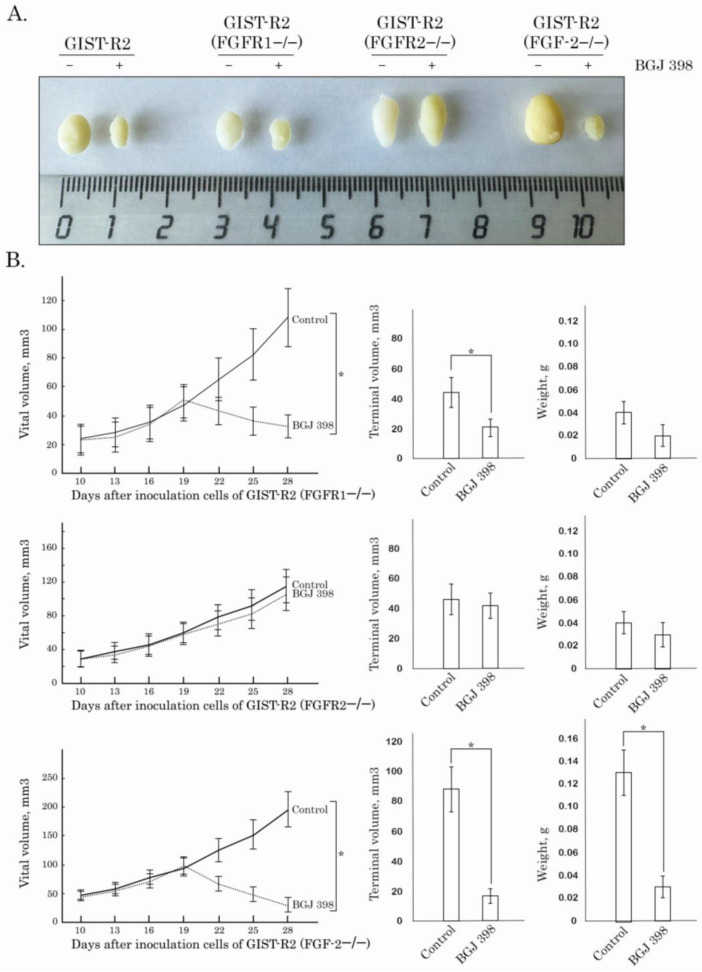
BGJ 398 exhibits anti-tumor activity via FGFR2-signaling axis. After subcutaneous inoculation of FGF-2−/−, FGFR1−/−, and FGFR2−/− cells (day 14), nude mice were randomized into 4 groups (n = 4) and administered i.p. 100 μL of vehicle (negative control) or BGJ 398 (20 mg/kg). The changes in tumor sizes were calculated as a percentage of the baseline. (**A**) Representative images of the final tumor volumes in each experimental group. (**B**) Dynamics of the tumors growth, tumors volumes and weights for each experimental group. * *p* < 0.001.

**Table 1 cancers-15-05354-t001:** An increased sensitivity of GIST-R2 cells to BGJ 398.

Target Drug	GIST-R1	GIST-R2	Fold Increase
IM	47.1 µM ± 1.4	17.8 µM ± 0.8	2.65
BGJ 398	4.4 µM ± 0.2	<0.0003 µM	>14,700

## Data Availability

The data and materials shown in the present manuscript are available from the corresponding authors upon reasonable request.

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
