# Peer review of "Unraveling the Mechanisms of Sensitivity to Anti-FGF Therapies in Imatinib-Resistant Gastrointestinal Stromal Tumors (GIST) Lacking Secondary KIT Mutations"

_cancers, 2023, doi:10.3390/cancers15225354_

Round 1

Reviewer 1 Report

Comments and Suggestions for Authors

This study evaluated the primary signaling pathways and potential mechanisms underlying a subpopulation of GIST cells resistant to IM. They conducted a comprehensive investigation, employing various experimental techniques, to assess the effects of FGF1/2 and FGFR1/2 expression on the sensitivity of GIST cells lacking secondary KIT mutation. Their findings support the inclusion of FGFR inhibitors as a therapeutic approach to enhance IM sensitivity in these cells.

In general,  this research has offered a valuable therapeutic approach in the field of cancer treatment, underscoring its potential importance for clinical practice. I have just one significant question regarding GIST-R2:

Both GIST-R1 and GIST-R2 have demonstrated resistance to IM, with GIST-R2 originating from GIST-R1 after a long-time IM treatment. I'm curious about the proportion and changes in the percentage of GIST-R2 within its parental population. Is GIST-R2 a minor subpopulation, or does its population steadily increase during treatment? Considering that the authors have picked single-cell clones to establish the GIST-R2 cell line (if my understanding is correct), are there any other subpopulations exhibiting similar behavior or possessing similar genomic/mutation characteristics but differing in sensitivity compared to GIST-R2?

Furthermore, I have two minor suggestions that could enhance the paper's clarity:

1. Enhance the citation of references within the paper. For instance, consider including references following the sentence: A long time ago, secondary KIT mutations in GIST were found as a predominant molecular mechanism of acquired IM resistance in GIST.

2. To facilitate readability, consider adding descriptions and captions for the supplementary figures.

Author Response

We thank very much the reviewer for the detailed analysis of our manuscript and comments and suggestions, as well. Below are our specific responses to reviewer’s comments (shown in quotes and italics). The changes in the revised version of the manuscript are highlighted with yellow.

  1. This study evaluated the primary signaling pathways and potential mechanisms underlying a subpopulation of GIST cells resistant to IM. They conducted a comprehensive investigation, employing various experimental techniques, to assess the effects of FGF1/2 and FGFR1/2 expression on the sensitivity of GIST cells lacking secondary KIT mutation. Their findings support the inclusion of FGFR inhibitors as a therapeutic approach to enhance IM sensitivity in these cells.

In general, this research has offered a valuable therapeutic approach in the field of cancer treatment, underscoring its potential importance for clinical practice. I have just one significant question regarding GIST-R2:

Both GIST-R1 and GIST-R2 have demonstrated resistance to IM, with GIST-R2 originating from GIST-R1 after a long-time IM treatment. I'm curious about the proportion and changes in the percentage of GIST-R2 within its parental population. Is GIST-R2 a minor subpopulation, or does its population steadily increase during treatment? Considering that the authors have picked single-cell clones to establish the GIST-R2 cell line (if my understanding is correct), are there any other subpopulations exhibiting similar behavior or possessing similar genomic/mutation characteristics but differing in sensitivity compared to GIST-R2?” 

We greatly appreciate reviewer for these comments and for raising up very important point about the proportion of GIST-R2 cells raised up after the long-term inhibition of KIT-signaling in GIST-R1 cells. To clear this point, we included a new Figure in revised manuscript, illustrating that ~20% of GIST-R2 cells lacked c-Kit expression (Supplementary Figure 2). We assume that these cells represent population of GIST-R2 cells since c-KIT expression was found to be significantly reduced in GIST-R2 cells when compared with GIST-R1 and naïve GIST-R1 cells. This data is shown in Figure 8A. The proliferation speed of GIST-R2 cells looks a bit higher when compared with R1 cells, as shown in Figure 5. Therefore, we expect an increase of the amount of GIST-R2 cells during a long-term culture. But this is just a proposal and we did not examine this possibility precisely.

Furthermore, I have two minor suggestions that could enhance the paper's clarity:

“1. Enhance the citation of references within the paper. For instance, consider including references following the sentence: A long time ago, secondary KIT mutations in GIST were found as a predominant molecular mechanism of acquired IM resistance in GIST”. We appreciate reviewer for this comment and suggestion and included the references 6-8 illustrating the role of secondary KIT mutations in GIST resistance to IM.  

“2. To facilitate readability, consider adding descriptions and captions for the supplementary figures”. Thank you very much for this suggestion. We added descriptions and captions for the supplementary figures.

Reviewer 2 Report

Comments and Suggestions for Authors

Comments:

1. Is R1 vs R2 different on morphology? please show higher power image.

2. Any p53 data on R1 vs R2 cells?

3. On figure3, why STAT1 disappeared in BGJ and IM+BGJ groups but phospho-STAT1 appeared? Would antibody against STAT1 recognizes STAT1 and p-STAT1 both?

4. Any data on S6-Ribosomal protein (total)?

5. I do not understand the Figure 8C. The pixel density is from what part of Figure B? Please explain on legend.

Author Response

We thank very much the reviewer for the detailed analysis of our manuscript and comments and suggestions, as well. Below are our specific responses to reviewer’s comments (shown in quotes and italics). The changes in the revised version of the manuscript are highlighted with yellow.

Comments:

“1. Is R1 vs R2 different on morphology? please show higher power image.”  We appreciate reviewer for this comment. The morphology of R1 and R2 cells is similar (e.g. spindle-shape), which is shown in the Supplementary Figure 1A.  

“2. Any p53 data on R1 vs R2 cells?” R1 and R2 cells were generated from GIST T-1 cells which are known to be p53 negative. We also revealed this point illustrating no changes in p53 status in R1 and R2 cells by NGS as was highlighted in chapter 3.2. in revised manuscript.

“3. On figure3, why STAT1 disappeared in BGJ and IM+BGJ groups but phospho-STAT1 appeared? Would antibody against STAT1 recognizes STAT1 and p-STAT1 both?”  We greatly appreciate a reviewer for this notice. We re-run WB and replaced the image for both total and phosphorylated forms of STAT-1, as shown in Figure 3. the Abs are not cross-reacting with each other and specifically recognizing the total and phosphorylated forms of this protein.

“4. Any data on S6-Ribosomal protein (total)?”  We greatly appreciate a reviewer for this notice, too. Unfortunately, we do not have commercial Abs recognizing total S6-Ribosomal protein during the preparation of our manuscript. But we totally agree with reviewer’s comment about an importance to show both forms of the proteins. In this case, we can also exclude phosphorylated form of S6 from this figure.

  1. I do not understand the Figure 8C. The pixel density is from what part of Figure B? Please explain on legend. Yes, it is. It is currently explained in the legend in revised manuscript.

Round 2

Reviewer 2 Report

Comments and Suggestions for Authors

No more comments